# Real-World Clinical Outcomes Associated with Canagliflozin in Patients Aged 65 Years and Older with Type 2 Diabetes Mellitus in Spain: The Old Real-Wecan Study

**Manuel A. Gargallo-Fernández** [1,2,*], **Alba Galdón Sanz-Pastor** [2], **Teresa Antón-Bravo** [3], **Miguel Brito-Sanfiel** [4], **Jaime Wong-Cruz** [5] and **Juan J. Gorgojo-Martínez** [5]

1   Department of Endocrinology and Nutrition, Hospital Universitario Infanta Leonor, 28031 Madrid, Spain
2   Department of Endocrinology and Nutrition, Hospital Fundacion Jiménez Díaz, 28040 Madrid, Spain; albagaldonsp@gmail.com
3   Department of Endocrinology and Nutrition, Hospital Universitario de Móstoles, Móstoles, 28935 Madrid, Spain; tantonb1978@gmail.com
4   Department of Endocrinology and Nutrition, Hospital Universitario Puerta de Hierro, Majadahonda, 28222 Madrid, Spain; mbritosanfiel@hotmail.com
5   Department of Endocrinology and Nutrition, Hospital Universitario Fundación Alcorcón, Alcorcón, 28922 Madrid, Spain; jemwong@fhalcorcon.es (J.W.-C.); juanjo.gorgojo@gmail.com (J.J.G.-M.)
*   Correspondence: gargallomgar@gmail.com

**Abstract:** The observational Real-Wecan study showed that canagliflozin 100 mg (CANA100) as an add-on therapy, and canagliflozin 300 (CANA300), switching from prior SGLT-2i therapy, significantly improved several cardiometabolic parameters in patients with T2DM. The aim of this sub-analysis was to assess the effectiveness and safety of canagliflozin in patients aged ≥65 years. The primary outcome of the study was the mean change in HbA1c over the follow-up period. A total of 583 patients met the inclusion criteria (39.5% >65 years), 279 in the cohort of CANA100 (36.9% >65 years; mean HbA1c 8.05%) and 304 in the cohort of CANA300 (mean age 61.1 years; 41.8% ≥65 years; mean HbA1c 7.51%). In the CANA100 cohort, older patients showed significant reductions in HbA1c (−0.78%) and weight (−4.5 kg). Patients aged ≥65 years switching to CANA300 experienced a significant decrease in HbA1c (−0.27%) and weight (−2.1 kg). There were no significant differences in HbA1c and weight reductions when the cohorts of patients <65 and ≥65 years were compared in a multiple linear regression model. The safety profile of canagliflozin was similar in both age groups. These findings support canagliflozin as an effective therapeutic option for older adults with T2DM.

**Keywords:** canagliflozin; SGLT-2 inhibitor; elderly patients; type 2 diabetes mellitus; real-world study

## 1. Introduction

Diabetes, particularly type 2 diabetes mellitus (T2DM), is becoming more prevalent in the general population, especially in individuals over the age of 65 years. Older patients with T2DM typically have a longer duration of disease, contributing to reduced pancreatic functioning, more comorbidities and greater medication complexity than those seen in younger people with T2DM. Hence, treatment of T2DM in older patients requires careful consideration of the increased risk of hypoglycaemia, cardiovascular and renal complications, frailty, and more serious drug-related adverse events (AE). Therefore, selection of any antihyperglycaemic agents (AHA) for treating these patients must consider the benefit/risk ratio. There are several current guidelines that specifically address the management of T2DM in this population to achieve glycaemic control with minimal risks [1–5].

Canagliflozin is one of the new therapies introduced in T2DM treatment in the last years. It belongs to the sodium–glucose co-transporter type 2 inhibitors (SGLT-2is) class of AHA drugs that reduce tubular reabsorption of filtered glucose, inducing urinary glucose excretion in individuals with hyperglycaemia, which leads to decreased plasma glucose

as well as an osmotic diuresis and net caloric loss [6–9]. In phase 3 studies, canagliflozin has been shown, in patients with T2DM, to improve glycaemic control and reduce body weight and blood pressure (BP) in combination with different AHA therapies [10–14]. In addition, canagliflozin has demonstrated cardiovascular (CV) and renal benefits in patients with T2DM and high CV risk or chronic kidney disease (CKD) [15,16].

However, in most of these studies, the mean age of the population was below 65 years, so it is questionable to extend these results to older patients. In addition, no randomised clinical trials (RCTs) or real-world studies (RWS) have evaluated the possible outcome differences between the two age groups with the strategy of intensification of the SGLT-2i therapy by switching to CANA300, either from CANA100 or other SGLT-2is.

In summary, there is scarce data from RWS regarding the effectiveness and safety of canagliflozin comparing younger and older patients, and no data of the intensification of SGLT-2is therapy by switching to CANA300 in elderly patients with T2DM.

We have previously studied in a real-world setting the effectiveness of CANA100 and the switch to CANA300 from prior SGLT-2is in patients with T2DM [17]. In that study, we found significant improvement of several cardiometabolic parameters in patients with T2DM in a real-world setting, with a low incidence of AEs and a high rate of persistence, confirming the results from phase 3 RCTs with CANA100 and obtaining real-life evidence on the effectiveness and safety of switching to CANA300 from a prior SGLT-2i therapy in patients with suboptimal metabolic control. We performed a subanalysis, the old-aged patients real-world evidence with canagliflozin (Old Real-Wecan), of the previously multicentric retrospective study, to assess in a real-world setting the effectiveness and safety of CANA100 as a daily add-on to the background antihyperglycaemic therapy, and to evaluate the intensification of SGLT-2i therapy, by switching to CANA300 daily, either from CANA100 or other SGLT-2is, in patients of 65 years and older compared to younger individuals.

## 2. Materials and Methods

This subgroup analysis was based on data from the Real-Wecan study previously described [17]. Briefly, the Real-Wecan study is a multicentric retrospective study whose aims were to assess in a real-world setting the effectiveness and safety of canagliflozin 100 mg/d (CANA100) as an add-on to the background AHAs, and to evaluate the intensification of prior SGLT-2i therapy by switching to canagliflozin 300 mg/d (CANA300) in patients with T2DM. The patients (older than 18 years) were consecutively selected from the diabetes clinic databases.

Therapy with CANA100 or CANA300 was started at least 6 months before data collection and patients received at least 1 dose of the drug.

In patients with suboptimal HbA1c or weight response prior to SGLT-2is, switching to CANA300 was performed.

### 2.1. Outcomes and Study Measures

All data were obtained from medical records. Three visits were recorded: V1, baseline 1 (CANA100 mg) or switch (CANA300); V2, 6 + 2 months after the start of CANA100 or after switching to CANA300; V3, last visit of the follow-up period.

Clinical parameters registered at baseline are shown in Table 1. At V1, V2, and V3: fasting plasma glucose (FPG), weight, Body Mass Index (BMI), systolic blood pressure (SBP), and diastolic blood pressure (DBP) were collected. Any death and all causes of withdrawals (WDs) were registered. AEs associated in clinical trials to SGLT-2is (Table 2) were collected at each visit.

**Table 1.** Baseline characteristics of the patient cohorts by age group with canagliflozin 100 mg (CANA100) and canagliflozin 300 mg (CANA300).

| | CANA100 | | | CANA300 | | |
|---|---|---|---|---|---|---|
| | <65 Years (n 176) | ≥65 Years (n 103) | *p* | <65 Years (n 177) | ≥65 Years (n 127) | *p* |
| Follow-up period (months) * | 9.1 (5.0–20.3) | 9.2 (5.9–22.3) | 0.402 | 17.5 (7.9–30.4) | 14.0 (6.7–24.3) | 0.013 |
| Gender (male/female) | 54/46 | 56.3/43.7 | 0.705 | 57.1/42.9 | 54.3/45.7 | 0.636 |
| Age (years) | 52.8 (10.0) | 71.5 (4.9) | <0.0001 | 54.5 (8.2) | 70.3 (3.5) | <0.0001 |
| Duration of T2DM (years) * | 8.0 (3.7–13.4) | 12.0 (8.4–17.8) | <0.0001 | 10.1 (5.8–15.2) | 15.9 (11.2–21.0) | <0.0001 |
| HbA1c (%) | 8.11 (1.60) | 7.94 (1.40) | 0.375 | 7.48 (1.24) | 7.55 (0.93) | 0.572 |
| Patients with HbA1c >7% | 74.0 | 76.2 | 0.677 | 67.2 | 76.4 | 0.083 |
| Fasting plasma glucose (mg/dL) | 167.1 (61.8) | 156.8 (49.1) | 0.131 | 144.4 (38.1) | 142.6 (32.9) | 0.664 |
| Weight (kg) | 99.2 (22.1) | 87.5 (16.9) | <0.0001 | 97.6 (22.1) | 84.2 (16.7) | <0.0001 |
| BMI (kg/m$^2$) | 35.8 (7.2) | 33.1 (6.8) | 0.003 | 35.6 (7.7) | 32.0 (6.9) | 0.002 |
| SBP (mmHg) | 137.8 (20.3) | 139.7 (20.0) | 0.485 | 136.3 (13.7) | 135.2 (15.6) | 0.560 |
| DBP (mmHg) | 80.4 (10.0) | 77.3 (10.0) | 0.018 | 79.8 (10.8) | 73.6 (10.2) | <0.0001 |
| eGFR (mL/min/1.73 m$^2$) | 92.4 (14.3) | 76.6 (15.0) | <0.0001 | 90.4 (16.6) | 77.5 (14.5) | <0.0001 |
| Microalbuminuria (mg/g Cr) * | 6.0 (4.0–31.1) | 9.8 (4.3–37.0) | 0.417 | 7.2 (0–25.4) | 8.8(0–27.6) | 0.715 |
| Diabetic renal disease (%) | | | | | | |
| Stage G0/G1 | 67.9 | 24.7 | <0.0001 | 62.9 | 27.6 | <0.0001 |
| Stage G2 | 30.9 | 56.7 | | 32.9 | 56.7 | |
| Stage G3a | 1.2 | 17.5 | | 3.0 | 15.7 | |
| Stage G3b | 0 | 1.0 | | 0.6 | 0 | |
| Stage G4–G5 | 0 | 0 | | 0.6 | 0 | |
| Stage A1 | 73.9 | 75.0 | 0.179 | 80.0 | 78.4 | 0.773 |
| Stage A2 | 22.8 | 15.6 | | 16.8 | 19.6 | |
| Stage A3 | 3.3 | 9.4 | | 3.2 | 2.1 | |
| Hypertension | 68.2 | 82.5 | 0.009 | 81.9 | 85.8 | 0.365 |
| Hypercholesterolaemia | 75.0 | 88.3 | 0.007 | 87.6 | 95.3 | 0.022 |
| Hypertriglyceridaemia | 44.6 | 31.1 | 0.026 | 56.5 | 45.2 | 0.053 |
| Combined hyperlipidaemia | 38.3 | 31.1 | 0.225 | 51.4 | 45.2 | 0.289 |
| Diabetic renal disease | 18.8 | 24.3 | 0.273 | 26.6 | 29.1 | 0.62 |
| Diabetic neuropathy | 7.4 | 8.7 | 0.686 | 8.5 | 15.7 | 0.05 |
| Coronary artery disease | 8.0 | 9.8 | 0.597 | 5.6 | 17.3 | 0.001 |
| Stroke | 2.8 | 5.8 | 0.216 | 3.4 | 3.9 | 0.801 |
| Diabetic Retinopathy | 13.1 | 13.6 | 0.915 | 16.4 | 21.3 | 0.279 |
| Peripheral artery disease | 1.7 | 5.8 | 0.06 | 6.8 | 12.6 | 0.084 |

**Table 1.** *Cont.*

| | CANA100 | | | CANA300 | | |
|---|---|---|---|---|---|---|
| | **<65 Years (n 176)** | **≥65 Years (n 103)** | *p* | **<65 Years (n 177)** | **≥65 Years (n 127)** | *p* |
| Arrhythmias | 1.1 | 10.7 | <0.0001 | 4.0 | 5.5 | 0.523 |
| Heart failure | 1.1 | 2.9 | 0.280 | 2.8 | 3.1 | 0.869 |
| Antihyperglycaemic agent: | | | | | | |
| Metformin | 81.3 | 85.4 | 0.371 | 87.6 | 90.6 | 0.416 |
| Sulphonylureas or glinides | 15.9 | 23.3 | 0.126 | 11.9 | 7.1 | 0.168 |
| Pioglitazone | 0.6 | 2.9 | 0.112 | 1.7 | 4.7 | 0.124 |
| DPP-4 inhibitors | 38.6 | 46.6 | 0.193 | 20.3 | 29.9 | 0.055 |
| GLP-1 receptor agonist | 34.7 | 28.2 | 0.262 | 63.3 | 58.3 | 0.377 |
| Insulin | 37.5 | 39.8 | 0.702 | 39.0 | 46.5 | 0.193 |
| SGLT-2i | 0 | 0 | | 100 | 100 | NA |
| Dapagliflozin 10 mg | | | | 47.7 | 49.6 | |
| Canagliflozin 100 mg | | | | 29.5 | 28.3 | |
| Empagliflozin 10 mg | | | | 5.1 | 9.4 | |
| Empagliflozin 25 mg | | | | 10.8 | 1 | |
| Antihypertensive drugs | 61.4 | 80.7 | 0.004 | 76.8 | 81.1 | 0.007 |
| Lipid-lowering drugs | 70.5 | 81.6 | 0.165 | 84.7 | 91.3 | 0.124 |

Data: percentage or mean (SD), except * median (IQR). BMI: Body Mass Index. DBP: diastolic blood pressure. eGFR: estimated glomerular filtrate rate. SBP: systolic blood pressure.

**Table 2.** Safety outcomes.

| | CANA100 | | CANA 300 | |
| --- | --- | --- | --- | --- |
| | < 65 Years (n 176) | ≥65 Years (n 103) | <65 Years (n 177) | ≥65 Years (n 127) |
| Withdrawals (%) * | 12 | 8.7 | 9 | 9.4 |
| Deaths (%) | 1 | 0 | 0 | 0 |
| AEs of special interest | | | | |
| Genital mycotic infections (%) | 13 | 9.7 | 10.2 | 7.9 |
| Urinary tract infections (%) | 4.5 | 4.9 | 5.6 | 9.4 |
| Hypoglycaemias (%) | 7.4 | 9.7 | 6.2 | 13.4 |
| Intravascular volume-related AEs (%) | 1.1 | 1.9 | 0 | 0.8 |
| Fractures (%) | 0.6 | 1.9 | 0 | 0 |
| Polycythaemia (%) | 1.7 | 0 | 1.1 | 0.8 |
| Ketoacidosis (%) | 0 | 0 | 0 | 0 |
| Amputations (%) | 0 | 0 | 0 | 0 |

Differences between age groups were not statistically significant, except for hypoglycaemias in the CANA300 cohort ($p < 0.033$). AEs: Adverse Events. * Withdrawals in the CANA100 cohort other than switching to CANA300.

The main outcomes were to assess changes in HbA1c from V1 to V2 and V3 in both cohorts of CANA100 and CANA300, and to compare these results between subgroups of participants younger than 65 and aged 65 and older. Secondary outcomes were to assess changes in weight, eGFR, microalbuminuria, lipid profile, BP, liver function tests, hematocrit, and serum uric acid, and to compare these results between subgroups of participants younger than 65 and aged 65 and older.

*2.2. Statistical Methods*

Results are shown as mean (standard deviation [SD]) (continuous variables that follow a normal distribution) or as median (interquartile range [IQR]) (those that do not meet normality criteria). Categorical data are shown in percentages. Analyses were carried out using the available data without any imputation of missing data. McNemar tests, Paired t-tests and Wilcoxon tests were performed to compare baseline data to that at follow-up. A multiple linear regression analysis was performed to assess the adjusted mean differences between older and younger patients, controlling for those baseline variables with statistically significant differences between both age groups.

All analyses were conducted by using 2-sided tests and a significance level of 0.05 with the Statistical Package for the Social Sciences (SPSS) version 15.0.1 (IBM Corp., Armonk, NY, USA).

**3. Results**

*3.1. Demographic and Baseline Characteristics*

A total of 583 patients met the inclusion criteria (230 (60.5%) ≥65 years, 33 (5.7%) >75 years), 279 in the cohort of CANA100 (men 54.8%, mean age 59.7 years, 36.9% >65 years, mean HbA1c 8.05%, mean BMI 34.8 kg/m$^2$) and 304 in the cohort of CANA300 (men 55.9%, mean age 61.1 years, 41.8% ≥65 years, mean HbA1c 7.51%, mean BMI 34.5 kg/m$^2$). Baseline characteristics of the patients from different age groups and canagliflozin dose groups are shown in Table 1.

In both cohorts, patients aged ≥65 years showed a longer T2DM duration, lower BMI, higher prevalence of high BP (in the CANA100 cohort) and hyperlipidaemias, as a more advanced chronic kidney disease (CKD).

*3.2. Analyses of Effectiveness*

In the CANA100 cohort (median follow-up 9.2 months), patients aged ≥65 years showed significant reductions in HbA1c at V2 and V3 (−0.70% [95% CI −0.44 to −0.93] and –0.78% [95% CI −0.55 to −1], respectively, both $p < 0.0001$), and the percentage of

patients with HbA1c below 7% significantly increased from 23.8% at V1 to 49.5% at V3 (*p* < 0.0001). Patients younger than 65 years showed significant reductions in HbA1c at V2 and V3 (−1.06% [95% CI −0.73 to −1.39] and −0.97% [95% CI −0.72 to −1.21] respectively, both *p* < 0.0001), and the percentage of patients with HbA1c below 7% significantly increased from 26% at V1 to 51.2% at V3 (*p* < 0.0001).

In the subgroup of patients aged ≥65 years with suboptimal glycaemic control, defined as baseline HbA1c > 7% (mean HbA1c 8.48%), CANA100 decreased HbA1c at V2 and V3 (−0.83% [95% CI −0.53 to −1.1] and –1.0% [95% CI −0.73 to −1.2] respectively, both *p* < 0.0001). In those patients with poor glycaemic control, defined as baseline HbA1c > 8% (mean HbA1c 9.1%) CANA100 decreased HbA1c at V2 and V3 (−1% [95% CI −0.62 to −1.5] and –1.3% [95% CI −0.91 to −1.7] respectively, both *p* < 0.0001).

CANA100 was associated with a significant weight loss (WL) in older patients at V2 and V3 (−3.9 kg [95% CI −0.67 to −2.6] and −4.5 kg [95% CI −2.9 to −6] respectively, both *p* < 0.0001). In younger patients, CANA100 was associated with a significant WL at V2 and V3 (−3.62 kg [95% CI −2.62 to −4.63] and −3.79 kg [95% CI −2.87 to –4.71] respectively, both *p* < 0.0001).

In addition, in the CANA100 cohort, older patients showed a non-significant reduction in SBP (–4.6 mmHg) and a significant decrease in DBP at V3 (−2.4 mm Hg [95% CI −0.23 to −4.5], *p* < 0.05).

In those individuals with SBP > 140 mmHg at V1 (mean SBP 155 mmHg), CANA100 lowered SBP levels at V3 by −16.1 and −14.1 mmHg in older and younger patients, respectively (both *p* < 0.0001).

In the multivariate analysis, after controlling for duration of T2DM, estimated glomerular filtration rate (eGFR), weight, A1c, HTA and dyslipidaemia, and adjusted differences in HbA1c reduction, WL and BP changes between both age groups were not statistically significant.

In the CANA300 cohort (median follow-up 14 months), patients aged ≥65 years showed, starting from a median HbA1c 7.58% at V1, significant reductions in HbA1c at V2 and V3 (−0.36% [95% CI −0.04 to −0.17] and –0.27% [95% CI −0.07 to −0.11] respectively, both *p* < 0.0001). The percentage of patients with HbA1c below 7% significantly increased from 23.6% at V1 to 40.7% at V3 (*p* < 0.0001). Patients younger than 65 years showed, starting from a median HbA1c 7.48% at V1, significant reductions in HbA1c at V2 and V3 (−0.51% [95% CI −0.34 to −0.68] and –0.40% [95% CI −0.22 to −0.58] respectively, both *p* < 0.0001). The percentage of patients with HbA1c below 7% significantly increased from 32.8% at V1 to 50.9% at V3 (*p* < 0.0001).

In the subgroup of patients aged ≥65 years with suboptimal glycaemic control, defined as baseline HbA1c > 7% (mean HbA1c 8.1%), CANA300 significantly decreased HbA1c at V2 and V3 (−0.42% [95% CI −0.20 to −0.64] and –0.39% [95% CI −0.20 to −0.58] respectively, both *p* < 0.0001). In those patients with poor glycaemic control, defined as baseline HbA1c > 8% (mean HbA1c 9.13%), CANA300 significantly lowered HbA1c at V2 and V3 (−0.92% [95% CI −0.51 to −1.3] and –0.91% [95% CI −0.53 to −1.3] respectively, both *p* < 0.0001). In the multivariate analysis, the adjusted differences in HbA1c reduction between both age groups were not statistically significant.

CANA300 was associated with significant WL in the older patient group at V2 and V3 (−2 kg [95% CI −1.1 to −2.8] and 2.1 kg [95% CI −1.1 to −3] respectively, both *p* < 0.0001). In younger patients, CANA300 significantly lowered the body weight at V2 and V3 (−2.14 kg [95% CI −1.41 to −2.87] and −2.15 kg [95% CI −1.18 to −3.13] respectively, both *p* < 0.0001).

There was a numerical reduction in SBP (−3.0 mmHg) and DBP (−1.8 mmHg) at the end of the follow-up period in older patients, without reaching statistical significance. In those patients with SBP > 140 mmHg at V1 (mean SBP 153 mmHg), CANA300 lowered SBP levels at V3 by −15.4 and −16.2 mmHg in older and younger patients, respectively (both *p* < 0.0001).

In the multivariate analysis, after controlling for duration of T2DM, eGFR, weight, A1c, HTA and dyslipidaemia, smoking and coronary heart disease, and the adjusted

differences in HbA1c reduction, weight loss and BP changes between both age groups were not statistically significant.

### 3.3. Analyses of Safety

In the CANA100 cohort, drug discontinuation was 36.9% and 48.9% in older and younger patients, respectively, over the entire follow-up period (Table 2). The main reason leading to CANA100 discontinuation in both age groups was switching to CANA300: 28.2% in older and 36.9% in younger patients (76.3% and 75.6%, respectively, of the whole discontinuation in both groups). Other reasons were GMIs (2.9%), UTIs (1.0%) and the worsening of kidney function (1.0%). No significant differences in withdrawals were observed between both age groups.

In the older age group, the most common AEs with CANA100 were GMI (9.7%), UTI (4.9%), mild hypoglycaemia (9.7%), intravascular volume-related AEs (1.9%), and fractures (1.9%), without significant differences with younger patients. No amputations, polyglobulia or ketoacidosis were reported.

In the CANA300 cohort, drug discontinuation rates were similar in both age groups (9.4% and 9.0% in older and younger patients, respectively) (Table 2). The main AEs leading to WD of CANA300 in patients aged ≥65 years were UTIs (2.4%), GMIs (0.8%), and the worsening of kidney function (0.8%).

A total of 7.9% of patients ≥65 years experienced GMIs, 9.4% experienced UTI, 13.4% experienced mild hypoglycaemia, 0.8% experienced polycythaemia, and 0.8% experienced intravascular volume-related AEs, without significant differences with younger patients, except a higher frequency of hypoglycaemias. No fractures, ketoacidosis or amputations were reported.

## 4. Discussion

Findings from this observational, retrospective, multicentre study show that CANA100 improved glycaemic control and reduced body weight and BP in patients with T2DM younger than 65, and aged 65 and older, without statistically significant differences between both age groups. In addition, switching to CANA300 from prior CANA100 or other SGLT-2is is associated with statistically significant HbA1c and weight reduction and improved SBP, regardless of age groups.

Regarding the effects of canagliflozin in older adults with T2DM, there is a RCT in individuals with T2DM aged 55 to 80 years comparing canagliflozin to a placebo [18], and a longer 78-week extension report of this same study [19]. Both of them showed that canagliflozin improved glycaemic control, reduced body weight and SBP, and was generally well tolerated. However, there were no data about possible differences with younger patients.

There are also some pooled analyses of 26-week, placebo-controlled phase 3 RCTs with canagliflozin, comparing individuals aged 65 years and older with younger patients [20,21], showing a similar decrease in HbA1c and BMI, although smaller HbA1c reductions in older participants with baseline eGFR 45 to <60 mL/min/m$^2$. In patients aged 75 and older, a pooled analysis of placebo-controlled randomised phase 3 studies of 18–26 weeks of duration found a lower numerical reduction in HbA1c efficacy of canagliflozin in older participants [22], possibly related to their lower mean baseline eGFR and a higher incidence of AEs.

While RCTs report clinical outcomes in controlled settings, such results may not be generalisable to patients seen in clinical practice, and RW, although statistically less rigorous, can provide valuable insight into how AHA perform within specific subgroups (such as older patients) often excluded in RCTs. We have identified some retrospective RWS with canagliflozin in older patients (≥65 years) [23–26], but in most of them the study design does not allow for the comparison of clinical outcomes with younger patients, except the study by Johnson et al. [25], which found smaller HbA1c reductions in older patients. Those results need to be interpreted with caution, due to the small sample size of

the older subgroup (66 patients) and the short follow-up period (mean time to last visit 215 days).

There are also some subanalyses from cardiovascular RCTs with empagliflozin and dapagliflozin [27,28] which show the cardio-renal benefits and safety of these SGLT-2is, regardless of age.

Our study compared two age groups (230 patients in the older cohort) from the same population with a median follow-up of around a year, in a real setting. Outcomes of our study show that the improvements with CANA100 therapy in glucose control, BP and weight previously described in this same population of patients with T2DM [17], in accordance with other RCT or RWS, are kept in older patients. Even the observation of greater reductions in HbA1c seen among those patients receiving canagliflozin, who had a higher baseline HbA1c, is a feature independent of age.

The results of our study in relation to changes in HbA1c (−0.78%) are in accordance with those described in RWS and pooled analysis in older patients with canagliflozin therapy [20–26]. Although younger patients showed numerically higher reduction in HbA1c than older ones, this was not a statistically significant difference, in contrast to the report from Johnson et al. [25] with a smaller sample size and shorter follow-up period than our study.

As previously mentioned, [22] reported data about patients aged 75 years and older. In our population, this subset of patients was only 5.7% and the comparison with younger patients did not show any statistically significant result.

We found several differences in baseline characteristics between age groups in both cohorts, such as a longer duration of T2DM and, therefore, more comorbidities as high BP (in the CANA100 cohort), CKD, heart disease or hyperlipidaemia are not surprising. On the other hand, the observed lower BMI is a common finding in people of advanced age for several reasons (loss of muscle mass, reduced energy intake, etc.). In summary, there were not any unexpected baseline differences between age groups.

Canagliflozin-reduced efficacy occasionally reported in the elderly may be a consequence of an age-dependent decline in estimated glomerular filtration rate (eGFR) rather than ageing per se, as has been described in some studies [22,29]. Pooled analyses by Gilbert et al. [21] only observed HbA1c reduction differences between age groups in the subset of participants with baseline eGFR 45 to <60 mL/min/1.73 m$^2$. In our study, despite a lower baseline eGFR in older patients, we did not observe any difference between age groups improving glycaemic control.

In relation to the strategy of intensification of CANA300 from other SGLT-2is not previously tested in this age group, we observed an additional HbA1c and weight reduction, raising the percentage of patients in good glycaemic control (HbA1c below 7%) from 23.6% to 40.7%, without age-related differences. In this case, improvement in HbA1c and weight was modest compared to that observed in the CANA100 cohort, but we should take into account that significant reductions in HbA1c and weight had already been attained with prior SGLT-2is therapy before the switch. In fact, HbA1c at V1 in the CANA300 cohort was 7.58%.

A reduction of SBP has been previously reported in older patients in canagliflozin therapy [15,16,18–22]. In our study, this effect was more relevant in the subset of patients with bad blood pressure control (SBP > 140 mm Hg). In this setting, both cohorts (CANA100 and CANA300) showed a statistically significant reduction, above 15 mm Hg in older patients, contributing to adequate blood pressure control. Larger reductions in DBP observed in the CANA100 cohort in older versus younger patients may be explained by more volume depletion in older patients [21,25].

The safety analysis was consistent with the findings observed in other RCTs [30] or RWS [31], and similar to the results of previous clinical studies of canagliflozin in elderly patients with T2DM [26]. The overall incidence of AEs with canagliflozin in our study was low, with most being mild or moderate in severity, but without differences between the two age groups. In fact, drug discontinuation rates were numerically more frequent in younger

than older patients in the CANA100 cohort. A higher rate of non-severe hypoglycaemia in older patients in the CANA300 cohort may be in relation to numerically more people being treated with insulin in this group. Nevertheless, special attention is needed to prevent hypoglycaemia in elderly patients, because severe hypoglycaemia is more likely to occur in these patients. An appropriate dose adjustment of insulin preparations and/or insulin secretagogues should be considered to prevent hypoglycaemia when using these drugs concomitantly with SGLT-2is.

Limitations of this study included the retrospective study design, which did not allow for control of the sample bias, but we have compared the results with the younger patients from the same sample. In addition, information on some variables was not recorded at each office visit, which resulted in missing data in some analyses. There is a potential recall bias in the frequency of AEs, as they were collected retrospectively. However, systematic research through electronic databases from primary care, laboratory departments and emergency departments was conducted to collect unreported AEs. The study design was not powered to find differences between age subgroups.

Additionally, this descriptive analysis did not control for other factors, such as the initiation of treatment with other medications, that might have contributed to the changes in body weight and BP observed over the course of the study. Since some AHAs were modified (either increased or decreased) over the follow-up period, especially in the CANA100 cohort, some influence of these changes on the final outcomes cannot be excluded.

It is also important to acknowledge that the results from this study may not be generalisable to broader populations because the sample in this study was selected from endocrinology clinics with generally more advanced T2DM (as disease duration and number of diabetes medications showed) than those from primary care settings.

Despite these limitations, the findings from the present study supplement the evidence on real-world outcomes with canagliflozin in the over-65 population with T2DM.

## 5. Conclusions

In summary, CANA100 (as an add-on therapy) and CANA300 (switching from CANA100 or other SGLT-2is) in patients with T2DM in a real-world setting showed similar effectiveness and safety, regardless of age. This subanalysis confirms the results of previous RCTs and RWS in older patients and adds real-life evidence on the strategy of switching to CANA300 from prior SGLT-2 therapy in that age group. Together, these findings support canagliflozin (CANA100 as an add-on therapy, or switching to CANA300 from other SGLT-2is) as a safe and effective therapeutic option for the increasingly frequent patients with T2DM older than 65 years or more, and contributes to clinicians feeling more confident to prescribe CANA to older patients with T2DM.

**Author Contributions:** All authors participated in the design, data collection, data interpretation, and critical review of the article. J.J.G.-M. performed the statistical analysis. M.A.G.-F. wrote the manuscript. All authors have read and agreed to the published version of the manuscript.

**Funding:** This research received no external funding.

**Institutional Review Board Statement:** The study was approved by the Ethical Review Boards (ethic code 19/56) of the centers which took part in the study and was performed in compliance with the ethical guidelines for research on humans. All the procedures were in accordance with the requirements set out in the international standards for epidemiological studies, as recorded in the International Guidelines for Ethical Review of Epidemiological Studies, and with the Helsinki Declaration of 1964, as revised in 2013. For this type of study, individual consent was not required.

**Informed Consent Statement:** Patient consent was waived due to retrospective data collection.

**Data Availability Statement:** The datasets generated during and/or analysed during the current study are available from the corresponding author on reasonable request.

**Conflicts of Interest:** M.A.G.-F. has the following financial relationships: lectures for Almirall SA, Astra-Zeneca, Boehringer Ingelheim Pharmaceuticals Inc., Janssen Pharmaceuticals, Eli Lilly and

Company, Novo-Nordisk, and Sanofi-Aventis; conducts research activities for Almirall SA, Astra-Zeneca, and Sanofi-Aventis. A.G.S.-P. has the following financial relationships: advisor on scientific boards for Astra-Zeneca and Janssen Pharmaceuticals; lectures for Astra-Zeneca, Boehringer Ingelheim Pharmaceuticals Inc., Janssen Pharmaceuticals, Eli Lilly and Company, Novo-Nordisk, Mundipharma Pharmaceuticals, Abbott, and Sanofi-Aventis. T.A.-B. has the following financial relationships: lectures for Astra-Zeneca, Mundipharma, Novo-Nordisk, Janssen Pharmaceuticals, Esteve, Eli Lilly and Company, and Sanofi-Aventis. M.B.-S. has the following financial relationships: advisor on scientific boards for Astra-Zeneca, Janssen Pharmaceuticals, Merck Sharp and Dohme, Novo-Nordisk, and Sanofi-Aventis; lectures for Almirall SA, Astra-Zeneca, Boehringer Ingelheim Pharmaceuticals Inc., Esteve, FAES, Eli Lilly and Company, Merck Sharp and Dohme, Mylan, Novo-Nordisk, and Sanofi-Aventis. J.J.G.-M. has the following financial relationships: advisor on scientific boards for Astra-Zeneca, Janssen Pharmaceuticals, Eli Lilly and Company, Merck Sharp and Dohme, and Novo-Nordisk; lectures for Abbott, AbbVie Inc., Astra-Zeneca, Boehringer Ingelheim Pharmaceuticals Inc., Esteve, Janssen Pharmaceuticals, Eli Lilly and Company, Merck Sharp and Dohme, Novo-Nordisk, Roche Pharma, and Sanofi-Aventis; conducts research activities for Astra-Zeneca and Sanofi-Aventis. J.W.-C. has no relevant financial interests to report.

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
