# Peer review of "Real-World Clinical Outcomes Associated with Canagliflozin in Patients Aged 65 Years and Older with Type 2 Diabetes Mellitus in Spain: The Old Real-Wecan Study"

_diabetology, doi:10.3390/diabetology2030015_

Round 1
Reviewer 1 Report
Congratulations on the article. Reflects the clinical reality of an efficient and effective therapeutic approach in people over 65 years of age.
Author Response
Thank you very much by your review
Reviewer 2 Report
This study was a subanalysis of the observational REAL WECAN study, aiming to assess the effectiveness and safety of canagliflozin (CANA) in patients aged >65 years. The primary outcome of the study was the mean change in HbA1c over the follow-up time. There were no significant differences in HbA1c and weight reductions when the cohorts of patients <65 and >65 years were compared in a multiple linear regression model. The safety profile of CANA was similar in both age groups. The authors concluded that these findings support CANA as an effective therapeutic option for older adults with T2DM.
The efficacy and safety of newer AHAs on elderly patients with T2DM in real-world study is important. The study was properly conducted and the manuscript was well-written. The conclusion was supported by the findings. There are only minor comments.
- Table 1. Please add “diabetic retinopathy”.
- Line 223. Drug discontinuation rates in older and younger groups (36.9 and 48.9%) seem to be high even in the real-world setting. The authors may comment on this.
Author Response
Thank you very much by your review
I have included diabetic rethinopathy in table I
I explain the seemingly high drug discontinuation rates in Results. More than 75% were siwtching to Canagliflozin 300
I Have checked the language
I submit the new manuscript (corecctions in red)
